# Pituitary–Adrenal Axis and Peripheral Immune Cell Profile in Long COVID

**DOI:** 10.3390/biomedicines12030581

**Published:** 2024-03-05

**Authors:** Jaume Alijotas-Reig, Ariadna Anunciacion-Llunell, Enrique Esteve-Valverde, Stephanie Morales-Pérez, Sergio Rivero-Santana, Jaume Trapé, Laura González-García, Domingo Ruiz, Joana Marques-Soares, Francesc Miro-Mur

**Affiliations:** 1Systemic Autoimmune Diseases Research Unit, Vall d’Hebron Institut de Recerca (VHIR), 08035 Barcelona, Spain; ariadna.anunciacion@vhir.org (A.A.-L.); joana.marquessoares@vallhebron.cat (J.M.-S.); 2Systemic Autoimmune Diseases Unit, Department of Internal Medicine, Hospital Universitari Vall d’Hebron (HUVH), 08035 Barcelona, Spain; 3Department of Medicine, Faculty of Medicine, Universitat Autònoma de Barcelona (UAB), 08035 Barcelona, Spain; doctor.esteve@gmail.com; 4Systemic Autoimmune Diseases Unit, Department of Internal Medicine, Hospital Universitari Parc Taulí, 08208 Sabadell, Spain; 5Systemic Autoimmune Disease Unit, Internal Medicine Department, Althaia Healthcare University Network of Manresa, 08243 Manresa, Spain; emoralesp@althaia.cat (S.M.-P.); srivero@althaia.cat (S.R.-S.); druizh@altahaia.cat (D.R.); 6Laboratory Medicine, Althaia Healthcare University Network of Manresa, 08243 Manresa, Spain; jtrape@althaia.cat (J.T.); lgonzalezg@althaia.cat (L.G.-G.); 7Tissue Repair and Regeneration Laboratory (TR2Lab), Institut de Recerca i Innovació en Ciències de la Vida i de la Salut a la Catalunya Central (IrisCC), 08500 Barcelona, Spain

**Keywords:** Long COVID, fatigue, dyspnea, cortisol, adrenal insufficiency, immune cells, inflammation

## Abstract

In Long COVID, dysfunction in the pituitary–adrenal axis and alterations in immune cells and inflammatory status are warned against. We performed a prospective study in a cohort of 42 patients who suffered COVID-19 at least 6 months before attending the Long COVID unit at Althaia Hospital. Based on Post-COVID Functional Status, 29 patients were diagnosed with Long COVID, while 13 were deemed as recovered. The hormones of the pituitary–adrenal axis, adrenocorticotropin stimulation test, and immune cell profiles and inflammatory markers were examined. Patients with Long COVID had significantly lower EuroQol and higher mMRC scores compared to the recovered individuals. Their symptoms included fatigue, myalgia, arthralgia, persistent coughing, a persistent sore throat, dyspnoea, a lack of concentration, and anxiety. We observed the physiological levels of cortisol and adrenocorticotropin in individuals with or without Long COVID. The results of the adrenocorticotropin stimulation test were similar between both groups. The absolute number of neutrophils was lower in the Long COVID patients compared to recovered individuals (*p* < 0.05). The total count of B lymphocytes remained consistent, but Long COVID patients had a higher percentage of mature B cells compared to recovered participants (*p* < 0.05) and exhibited a higher percentage of circulating resident memory CD8+ T cells (*p* < 0.05) and Treg-expressing exonucleases (*p* < 0.05). Our findings did not identify adrenal dysfunction related to Long COVID, nor an association between adrenal function and clinical symptoms. The data indicated a dysregulation in certain immune cells, pointing to immune activation. No overt hyperinflammation was observed in the Long COVID group.

## 1. Introduction

Individuals infected by SARS-CoV-2 may experience a constellation of long-lasting symptoms, including fatigue, myalgia, a sore throat, dyspnea, and coughing, with others including nervous system and neurocognitive disorders [1,2,3]. Similar long-lasting features were also observed in the SARS epidemic in 2003; one study reported that 17% of individuals infected with SARS-CoV-1 experienced long-term health issues one year after the infection [4], while others identified that symptoms resembling those of fibromyalgia were observed three years post-infection [5]. Researchers describe that 38% of SARS survivors encountered reduced lung oxygen flow 15 years after the initial infection [6]. During the recent pandemic, a group of COVID-19 survivors has been observed to battle with such long-term consequences [7]. Initially, physicians did not readily associate them with post-COVID-19 effects [8]. Despite uncertainties regarding attributing some cases to SARS-CoV-2 infection, recent data highlight the secondary effects of coronavirus infection, referring to them as Long COVID, persistent COVID, or post-acute COVID. In both the peer-reviewed literature [9] and public discussion [10], persistent symptoms have been reported among COVID-19 survivors, including those who initially experienced a mild acute illness [11,12,13,14]. Indeed, a study of Israeli healthcare workers underscored the Long-COVID risk following a breakthrough infection, even in fully vaccinated people [15]. These studies were conducted before the emergence of the dominant Omicron variants, which have been observed to decrease post-acute COVID symptoms [16]. Subsequently, treatment with Paxlovid in the acute phase of the infection has also been shown to be effective in reducing the risk of suffering post-acute COVID, regardless of whether patients were previously vaccinated or not [17]. Although vaccination before infection confers partial protection against Long COVID [18], the understanding that COVID-19 may extend beyond a transient respiratory disease and manifest as neurological and physical symptoms months after the initial infection, thereby increasing the overall burden of the disease, is gaining recognition among physicians [19].

The need to assess the multidimensional impact of certain conditions, including COVID-19, on patient health and quality of life using standardized scales is frequently established. Tools such as the Modified Medical Research Council (mMRC) Dyspnea Scale and the EuroQol (EQ)-5 Dimension (5D) or visual analogue scale (VAS) have been applied in previous studies to quantify symptom severity and impact on daily living, capturing the subtleties of the disease’s long-term effects, which may elude more immediate clinical assessments [20,21,22]. This nuanced understanding of post-infection symptoms is especially pertinent as we consider demographic vulnerabilities. While older patients may be at a higher risk for severe disease and death, younger survivors have also reported persistent symptoms weeks or months after acute illness [23].

Efforts to characterise the aetiology and pathophysiology of the late sequelae are ongoing and may reveal organ damage sustained during the acute infection phase [24]. This post-acute viral phase, during which individuals test negative for SARS-CoV-2, is sometimes persistent and is hypothesised to be associated with the residual or mild hypercytokinemia or dysfunction of the neuro-suprarenal axis, with accompanying subclinical adrenal hypofunction [25,26]. The impairment of the autonomic nervous system through disruption of tryptophan metabolism cannot be excluded [27,28]. This situation may or may not ease over time. However, in some patients, fatigue and other non-specific symptoms may last for more than six months, leading to the case being designated as viral chronic fatigue syndrome (ME/CFS). ME/CFS is a complex and poorly understood condition. Some studies evidenced a clear increased risk of developing ME/CFS in people who have had COVID-19 compared with those who have not [2,29]. Although many people first exhibit symptoms following a viral infection, the exact causes of this syndrome remain to be elucidated. One potential cause could be mitochondrial dysfunction [30]. Some immune cells in ME/CFS patients, particularly CD8+ T cells, show disruptions in energy production and use [31]. Additionally, both CD8+ and CD4+ T cells from ME/CFS demonstrate reduced glycolysis after activation. This diminished metabolism is inversely correlated with inflammatory cytokines in CD8+ T cells in patients affected by ME/CFS [31]. Severe COVID-19 may also induce long-term changes in the innate immune system through epigenetic modifications [32].

This study explored the premise that Long COVID symptoms are linked to pituitary–adrenal axis disruption, concomitant with a hyperinflammatory state and immune dysregulation. This hypothesis was assessed in a cohort of patients who had tested positive for SARS-CoV-2 at least six months prior to the enrolment. This work ultimately aims to contribute to the ongoing efforts to understand the underlying causes and mechanisms of Long COVID, potentially guiding more tailored management strategies for affected individuals.

## 2. Materials and Methods

### 2.1. Inclusion Criteria for Patients

Patients attending the Long COVID unit consultation at “Fundació Althaia Xarxa Assistencial” in Manresa (Barcelona, Spain) were invited to participate in this study, which was approved by the local ethics committee (CEI 21/82) and conducted according to the Declaration of Helsinki. All participants provided signed informed consent. Inclusion criteria included being between 18 and 70 years old and recording a positive PCR test for SARS-CoV-2 infection more than six months prior. Three expert clinicians (EEV, SMP, and SRS) from the Long COVID unit employed visual and verbal screening based on Post-COVID Functional Status (PCFS) [33] to determine if patients could be classified as having Long COVID. Enrolled patients were asked to answer PCFS questions regarding symptoms, pain, depression, anxiety, and their ability to perform household duties or activities independently. Those responding affirmatively were classified as having Long COVID. None of the enrolled patients indicated being unable to live alone without assistance. Blood samples collected for clinical management were analysed to assess leukocyte populations via flow cytometry, and basic blood tests and standard biochemistry parameters were also examined. None of the enrolled participants were treated with glucocorticoids during the Long COVID phase of the study. Demographic, clinical, and laboratory data were compiled in a database and stored in our institutional repository.

### 2.2. Health-Related Quality-of-Life Tests

The standardised health-related quality-of-life instrument EQ-5D assessed individuals’ overall health status. This generic tool quantitatively measures a person’s health and well-being, allowing for comparisons across different health conditions and populations. The EQ-5D comprises five health dimensions, namely, mobility, self-care, usual activities, pain/discomfort, and anxiety/depression, with each dimension having three levels (1 = no problem, 2 = moderate problems, and 3 = severe problems). In our Catalan cohort of patients, EQ-5D was assessed by using coefficients reported by [34], yielding an EQ-5D score ranging from 0 (worst health state) to 1 (best health state).

The mMRC dyspnoea scale was used to assess the severity of breathlessness, which is common in Long COVID patients. This scale uses a patient-reported measure of the impairment of daily life caused by breathlessness, which can range from minor discomfort to a factor that severely limits daily life. The scale ranges from grade 0, where patients experience no breathlessness, even with regular physical activity, to grade 4, where they report severe breathlessness that restricts them in their home or makes them short of breath when getting dressed or undressed. The intermediate stages are defined as follows: Grade 1, mild breathlessness during physical activity; Grade 2, shortness of breath when walking at an average pace; and Grade 3, having to pause for breath after walking for a few minutes.

As part of the assessment of the wide-ranging impact of the condition, the Long COVID Score (LCS) was formulated as LCS = mMRC/α + absolute value [log(EQ-5D + ε)], where the scaling factor α = 4. This balances the quantitative assessment of breathlessness with the multi-faceted evaluation of overall health and well-being, thus taking into account a broader range of health-related quality-of-life factors than just the respiratory system. The smoothing coefficient ε = 0.0758 is employed to guarantee that the resulting EQ-5D value is greater than 0 and allows for logarithm calculation.

### 2.3. Adrenal Function

Serum cortisol and adrenocorticotropin (ACTH) levels were assessed at baseline from blood samples drawn at 8:00 in the morning. The ACTH stimulation test was performed immediately via intramuscular injection of 0.25 mg of synthetic ACTH (Cigna Healthcare, Nashville, TN, USA) mixed with 2 mL of 0.9% sodium chloride. Blood was taken 60 min after ACTH injection. Cortisol levels in the serum were promptly assessed using an electrochemiluminescence immunoassay (ECLIA) with the Elecsys Cortisol II (Roche Diagnostics, Risch-Rotkreuz, Switzerland) in the Cobas e801 system (Roche Diagnostics). Basal serum ACTH levels were also determined via ECLIA with Elecsys ACTH (Roche Diagnostics) on the Cobas e801 system (Roche Diagnostics). A Δcortisol value represents the net increase in cortisol levels at 60 min post-ACTH injection, expressed as ΔCortisol = Cortisol 60′ − Basal cortisol.

### 2.4. Flow Cytometry

Blood samples drawn with EDTA for laboratory analysis were used to stain cells for flow cytometry. A 30-microliter aliquot of whole blood was stained with a 1:1 (*v*/*v*) mix of fluorescently conjugated antibodies to decipher leukocyte populations. This was performed in the presence of a human FcR blocking reagent (dilution 1:30; Milteny Biotech, Bergisch Gladbach, Germany) and Aqua Live/Dead cell fixable dye (1:1000; Molecular Probes, Eugene, OR, USA) to discard dead cells. The antibodies, used at dilution of 1:200, were anti-CD45-BV786 (HI30; BD Biosciences, Bergen, NJ, USA), anti-CD3-BB515 (UCHT1; BD Biosciences), anti-CD19-BV605 (HIB19; BioLegend, San Diego, CA, USA), anti-CD56-PE/Cy5 (B159; BD Biosciences), anti-γδTCR-PE (B1; BioLegend), anti-CD14-PE/CF594 (MfP9; BD Biosciences), anti-CD66b-APC/Cy7 (G10F5; BioLegend), anti-CD4-APC/R700 (RPA-T4; BD Biosciences), anti-CD8-BV650 (RPA-T8; BioLegend), anti-CD27-AF647 (M-T271; BioLegend), anti-CD45RA-PerCP/eF710 (GRT22; eBioscience, San Diego, CA, USA), anti-CD103-BV421 (Ber-ACT8; BioLegend), anti-αβTCR-BV650 (IP26; BD Biosciences), anti-CD73-BB515 (AD2; BD Biosciences), anti-CD38-PerCP/eF710 (HB7; eBioscience), anti-CD39-PE/CF594 (TU66; BD Biosciences), anti-HLA-DR-PE/Vio770 (REA805; Miltenyi Biotec), anti-CD25-AF647 (BC96; BioLegend), and anti-CD69-APC/Vio770 (REA824; Miltenyi Biotec). The antibody mixture was prepared in phosphate-buffered saline (PBS) containing 5 mM EDTA and 0.1% BSA (FACS buffer), along with a 1:10 (*v*/*v*) brilliant stain buffer (BD Biosciences). To determine the absolute cell number, 30 microliters of fluorescent beads (1000 beads/microliter; Beckman Coulter, Brea, CA, USA) were added to each tube. After a 30 min staining period, cells were fixed in 2% paraformaldehyde FACS lysing buffer (BD Biosciences) and washed with FACS buffer. Cellular data were acquired in a FACS LSRII (BD Biosciences) flow cytometer operated with FACS Diva software (BD Biosciences) and was analysed with FlowJo v10.6.0 (FlowJo-BD, Ashland, OR, USA).

### 2.5. Cytokine Assessment

Serum samples taken from participants were preserved at −80 °C and analysed collectively in a single batch. Concentrations of cytokines, including tumour necrosis factor (TNF)-α, interferon (IFN)-γ, interleukin (IL)-12p40, IL-12p70, IL-1β, IL-2, and IL-10, were assessed using the ProQuantum high-sensitivity immunoassay (Thermo Fisher Scientific, Waltham, MA, USA). This analysis was conducted on the QuantStudio 5 qPCR instrument (Thermo Fisher) utilising the ProQuantumTM Protein Biology software (Thermo Fisher). IL-6 serum concentration was measured via ECLIA with the Elecsys IL-6 reagent (Roche Diagnostics) on the Cobas e801 system (Roche Diagnostics). GDF-8 or myostatin was determined using an enzyme-linked immunosorbent assay (ELISA, Helsinki, Finland) kit (EH215RB, Invitrogen, Waltham, MA, USA), with readings taken on the Quanta-Lyser-2 plate reader (Werfen, San Diego, CA, USA).

### 2.6. Statistical Analysis

The Mann–Whitney U test, chi-square test, and Fisher’s exact were employed to compare differences between participants with or without Long COVID concerning quantitative and categorical variables, respectively. Continuous variables were expressed as median [interquartile range (IQR)], while categorical variables were presented as *n* (%). Two-way ANOVA was used to test for differences between groups, categorised by Long COVID status and sex. A two-sided α level of less than 0.05 was considered statistically significant. Statistical analysis was performed using the Rstats package in R software (v4.3.2).

## 3. Results

### 3.1. Long COVID Assessment

A total of 42 patients were enrolled for the study of Long COVID. The dates of their SARS-CoV-2 infection spanned from December 2020 to July 2021. These enrolled patients were part of the third wave of SARS-CoV-2 infections in Catalonia, Spain [35], which was attributed to an outbreak of the B.1.1.7 alpha variant [36] in this area.

The assessment of Long COVID took place, on average, 286 days after the primary infection, with a range of 217 to 346 days. Of the 42 enrolled individuals, 29 (69.05%) were diagnosed with Long COVID, while 13 (30.95%) were classified as fully recovered (Table 1). The time lapse between primary virus infection and the Long COVID assessment visit was comparable for individuals with Long COVID and those without Long COVID (268 [57] days vs. 303 [44.5] days, *p*-value = 0.0645).

### 3.2. Preclinical and Acute COVID-19 Profiles

Upon defining our two groups of participants, we examined differences in their preclinical history and the acute phase of COVID-19 (Table 1). The age of the participants was similar between both groups, whether diagnosed with Long COVID or not (53 [18] years old in Long COVID vs. 50 [8] years old in recovered participants, *p*-value = 0.64). Variables such as smoking, peripheral artery disease, previous arterial or venous thrombosis, active neoplasia, immunosuppressant treatments, HIV seropositivity, or chronic renal impairment were discarded as none of the enrolled patients presented these issues. There were no differences in the number of non-hospitalised individuals or the duration of hospitalisation between the two groups. The severity of COVID-19 among hospitalised patients was consistent between those with and without Long COVID. The treatments administered during the acute phase of COVID-19 and the medical complications observed were comparable between the group that later developed Long COVID and the group that fully recovered.

### 3.3. Symptom Evaluation and Quality of Life

Participants were evaluated using the mMRC scale to test for dyspnea related to activity, which ranges from 0 (no breathlessness) to 4 (severe breathlessness), and the EQ-5D questionnaire. In our cohort, breathlessness, which was measured with the mMRC scale (Figure 1A), demonstrated a significant association with Long COVID (*p*-value = 0.000123); additionally, Long COVID patients registered a lower score on the EQ-5D (Figure 1B) compared to those who had fully recovered (0.58 [0.19] vs. 1.00 [0.00], respectively; *p*-value = 0.000003).

Patients with Long COVID displayed a higher LCS than recovered individuals (0.84 [0.46] vs. 0.07 [0.00], respectively, *p*-value = 0.000000004) (Figure 1C). The above data suggested that Long COVID patients suffered more breathlessness and presented poorer quality of life than recovered individuals.

### 3.4. Vaccination and Seropositivity for SARS-CoV2, CMV and EBV Antibodies

Anti-SARS-CoV-2 vaccines were rolled out, starting in January 2021, with a prioritisation criterion in place. Our enrolled participants received their first shot, on average, 153 [31] days after their primary infection. At the time of their first Long COVID assessment, 88.24% of the participants had received at least one vaccine dose, and this distribution was similar among both Long COVID and fully recovered individuals (*p*-value = 0.55) (Appendix AA).

No differences were observed in the number of individuals testing seropositive for IgG anti-SARS-CoV-2 Spike protein or IgG anti-CMV between Long COVID and recovered groups (Appendix AB,C). Additionally, all participants were found to be IgG seropositive for anti-EBV. These data suggested that participants exhibited robust humoral immune responses.

### 3.5. Clinical Laboratory Parameters in Long COVID

We examined the biochemical and clinical laboratory parameters in patients with Long COVID to determine if there were any alterations compared to individuals who had fully recovered from COVID-19. Our analysis revealed that, on average, individuals with Long COVID had no dysfunctional laboratory parameters, similar to those who had recovered (Figure 2 and Appendix A).

### 3.6. Long COVID Symptomatology

In our cohort, we assessed the presence of the clinical symptoms related to Long COVID (Figure 3 and Appendix A). Our study identified several significantly more prevalent symptoms in patients diagnosed with Long COVID. These included fatigue, myalgia, arthralgia, dyspnea, persistent coughing, a persistent sore throat, anxiety, and a lack of concentration.

### 3.7. Pituitary–Adrenal Axis Function in Long COVID

The aforementioned symptoms, which are more closely linked to Long COVID, indicated the possibility of altered pituitary–adrenal axis function in individuals with this disorder. Therefore, we examined the cortisol levels in both groups (Figure 4A, left). Although cortisol levels in patients with Long COVID (10.0 [5.3] µg/dL) were lower than those observed in individuals without Long COVID (11.1 [5.8] µg/dL), no significant statistical difference was observed between both groups (*p*-value = 0.52). The reference values for cortisol at 8 a.m. varied between 6 and 18 µg/dL. Our analysis only detected evidence of hypocortisolaemia in the Long COVID patients, where two individuals (6.9%) displayed lower cortisol levels. Notably, this proportion corresponded to the prevalence of hypocortisolaemia in cases of ME/CFS. We observed hypercortisolaemia across both groups in our cohort in equal measure. We also investigated the correlation between cortisol levels and symptom severity, as evaluated using the LCS (Appendix A). The obtained results demonstrate no substantial correlation between cortisol levels and LCS (*p*-value = 0.98).

To determine whether Long COVID affected cortisol levels differently in women and men, we analysed cortisol variability, considering both Long COVID and sex (Figure 4A, right). The results of the two-way ANOVA statistical test indicate no significant differences (*p*-value = 0.26). However, it is noteworthy that the two cases of hypocortisolaemia were observed in female patients. In parallel with our observations on cortisol levels, we found that ACTH levels in patients with Long COVID (17.5 [11.7] µg/dL) were comparable to those without Long COVID (13.0 [4.7] µg/dL, *p*-value = 0.53) (Figure 4B).

Furthermore, we assessed the functional ability of the adrenal glands, focusing on the adrenal cortex, by conducting the ACTH stimulation test. As reflected in Figure 4C, the majority of the participants, comprising those with Long COVID, showed a typical response during the ACTH stimulation test. Following the intramuscular administration of ACTH (0.25 mg), cortisol levels exhibited significant escalation at 1 h. The rise in cortisol levels indicated that the adrenal glands were responding well to ACTH stimulation. The group with Long COVID experienced a comparable shift in cortisol levels (ΔCortisol = 15.0 [6.4] pg/dL) to that of the group without Long COVID (ΔCortisol = 16.1 [8.5] pg/dL, *p*-value = 0.58). The response pattern remained uniform when the data were examined by sex (Figure 4C, right). Nonetheless, a subset of individuals (*n* = 12, Appendix A) exhibited an inadequate cortisol response following ACTH administration. Adrenal insufficiency could be inferred when the cortisol levels increased by less than two-fold 60 min after ACTH stimulation (Appendix A). This inferior cortisol response was evenly distributed among individuals, regardless of their Long COVID status, and no statistically significant difference was observed (Fishers’ Exact test *p*-value = 0.72).

Regarding the symptoms described for Long COVID, those participants in our cohort exhibiting less than a two-fold cortisol induction in the ACTH stimulation test were found to be experiencing symptoms such as rhinitis, persistent coughing, and a persistent sore throat. However, there was no observed association between suboptimal cortisol response and depressive symptoms, anxiety, or fatigue (Appendix A). The same results were obtained when the analysis was solely restricted to the group of patients with Long COVID.

### 3.8. Immune Profile in Long COVID Patients

The methodology used for examining the immune cell populations in the participants of this study is explained in Appendix A, which outlines our gating strategy. We performed a comparative analysis to detect disparities in the number of assorted circulating immune cells between individuals who had been diagnosed with Long COVID and those who had completely recovered (Figure 5A and Appendix A).

The sole notable finding was that patients with Long COVID exhibited a lower number of circulating neutrophils in comparison with participants without Long COVID (2.6 [1.3] × 10^9^ Cells/L vs. 3.8 [0.7] × 10^9^ Cells/L, *p*-value = 0.0277, respectively). Nonetheless, their absolute neutrophil counts were still within the standard reference range (2–7 × 10^9^ Cells/L), precluding any clinical abnormality. No disparities were noted in the absolute tallies of overall T lymphocytes, T-helper lymphocytes, or cytotoxic lymphocytes. This uniformity was also evident in the tallies of memory and effector T cell populations within each lymphocyte subtype across both patient groups. Furthermore, the inquiry into the existence of HLA-DR+ CD38+ lymphocytes, which are indicative of viral infection, did not exhibit significant differences between the Long COVID and fully recovered cohorts. It is worth noting that the Long COVID group exhibited a higher proportion of Treg CD4+ HLA-DR+ CD38+ lymphocytes, although this disparity was not statistically significant (1.43% [0.55] vs. 1.11% [0.25], *p* = 0.08).

While the total B cell count was comparable between both groups, patients with Long COVID exhibited a higher proportion of mature B cells when compared to their recovered counterparts (Figure 5B and Appendix A). The median [IQR] was 25.8% [13.4] versus 17.9% [9.2], respectively (*p*-value = 0.0298). Long COVID patients exhibited a considerably greater proportion of CD8 T cells that expressed the CD103 integrin in circulation (2.8% [1.4] in Long COVID patients vs. 2.0% [0.5] in non-Long COVID patients, *p*-value = 0.033). This finding indicates more significant immune activation and an increase in the rate of resident memory CD8 cells in those experiencing Long COVID. The presence of regulatory T cells (Treg) was consistent in both groups. However, Long COVID patients had a greater proportion of Tregs expressing CD39 and CD73 markers (0.5% [1.1] vs. 0% [0], *p*= 0.027). No variations in other lymphocyte groups, including γδ-T cells, NK cells, and NK-T cells, were observed. In conclusion, our analysis suggests noticeable changes in particular lymphocyte subpopulations among patients with Long COVID, highlighting the intricate immune responses in these individuals.

### 3.9. Cytokine Evaluation

To investigate whether enduring immune cell activation is correlated to inflammation in Long COVID, we conducted tests on seven inflammatory markers in the sera of enrolled patients through ELISA (GDF-8), ECLIA (IL-6), or ProQuantum (TNF-α, IFN-γ, IL-1β, IL-2, IL-10, IL-12p40, and IL-12p70). Despite substantially higher levels of certain cytokines in some patients with Long COVID (Figure 6 and Appendix A), most of them did not show inflammation. Therefore, we did not discern a specific inflammatory cytokine with increased levels in patients with Long COVID relative to recovered individuals.

## 4. Discussion

In this preliminary investigation, we sought to analyse the functional physical status, adrenal function, and immune profiles of patients with Long COVID in comparison to individuals who had fully recovered from SARS-CoV-2 infection. Our study primarily ascertained that there was a noteworthy contrast in the EQ-5D score between Long COVID patients and those who had completely recovered. This disparity confirms that Long COVID patients experience a worsened quality of life, consistent with the persistent and debilitating nature of their symptoms. Furthermore, this study failed to identify any abnormalities in the pituitary–adrenal axis, particularly related to subclinical adrenal failure or insufficiency. One of the significant immunological findings in this study concerned B lymphocytes. Although the absolute number of B cells remained steady between the two groups, Long COVID patients showcased a higher ratio of mature B cells. These observations warrant certain comments that will be explained below.

The finding of a higher ratio of fully developed B cells in patients with Long COVID hints at potential alterations in B cell maturation or function in these patients, which could affect antibody production and immune response to reinfection or vaccination. Further investigation is needed to understand the significance of this finding in the context of Long COVID pathophysiology. In contrast, our study did not reveal significant differences in several other immune cell populations, including total T lymphocytes, T-helper cells, cytotoxic lymphocytes, and central and effector memory T cell subsets, between Long COVID patients and those who had fully recovered. This suggests that Long COVID may not be primarily characterised by overt changes in these particular immune cell populations, at least in the peripheral circulation. An intriguing finding was the higher percentage of resident memory CD8+ T cells in Long COVID patients. Resident memory T cells play a crucial role in immune surveillance at mucosal surfaces and may contribute to ongoing immune responses. The significance of this finding warrants further exploration, as it could provide insights into the persistence of symptoms in Long COVID. Another notable result was the higher percentage of Treg cells expressing exonucleases in Long COVID patients. Treg cells are essential for immune regulation and tolerance, and higher exonuclease enzyme surface expression could be indicative of Treg functional activation. This finding suggests the potential dysregulation of immune tolerance mechanisms, which may contribute to the chronic inflammatory state seen in Long COVID [25], but we failed to observe it. The measurement of a panel of cytokines in the plasma of all participants indicated signs of inflammation in some participants, regardless of having Long COVID or not, but mostly results remained within normal ranges. This finding is in line with another study [37] in which, despite describing the persistence of circulating SARS-CoV-2 spike antigens 12 months after infection, no inflammation was found in the tested cohort of Long COVID patients.

Cortisol is a hormone secreted by the adrenal glands in response to stress. Chronic illnesses, like Long COVID, can lead to persistent stress, resulting in the prolonged overstimulation of adrenal glands in order to maintain body balance. Over time, the adrenal cortex may fail to respond, despite the heightened activity of the hypothalamic–pituitary axis and the concurrent elevation of ACTH levels, leading to the development of subtle symptoms that may become apparent after a new stressor. Some clinical conditions align closely with these pathophysiological mechanisms. These include patients undergoing long-term corticosteroid treatment, cases of autoimmune adrenalitis, and those with post-viral chronic fatigue syndrome. Cortisol imbalance, i.e., slight adrenal hypofunction, can result in a range of symptoms. These may include fatigue, muscle and joint pain, feelings of weakness, mood disturbances, and cognitive deficits. Such symptoms are frequently cited by Long COVID patients, as previously reported [11,12,13] and our data agree. We assessed these distinct health conditions through various measures, including EQ-5D and mMRC. In a weighted LCS tool, we unified and equalised the scales. Our findings indicate that patients with Long COVID exhibited higher LCS than those who had recovered; however, these scores did not display any negative association with serum cortisol levels. Although we did not observe any dysfunction of the adrenal gland in our cohort of patients with Long COVID, two points should be highlighted. Firstly, we employed a standard and accepted method of stimulating the adrenal gland, which involved administering a 0.250 mg shot of ACTH and subsequently analysing the levels of blood cortisol over time. Studies argue against using this method to stimulate the pituitary–adrenal axis due to the supraphysiological doses of ACTH used [38,39]. There is a risk of misdiagnosing mild or early adrenal insufficiency [40]. Future studies using smaller ACTH doses should be conducted to analyse subclinical adrenal dysfunction. Secondly, 7% of patients in the Long COVID group exhibited slight hypocortisolemia. Nonetheless, these patients normalised their cortisol values after performing the ACTH stimulation test. For those patients suffering from subclinical adrenal insufficiency, the administration of small doses prednisone or 5-alpha-fluorocortisone may assist in their improvement [41]. Other researchers have examined the correlation between the adrenal gland, cortisol production, and Long COVID [25,26]. In their study, [25] demonstrated that lower cortisol levels were effective predictors of Long COVID status. However, the study did not disclose the occurrence of hypocortisolaemia, despite reporting lower cortisol levels in patients with Long COVID. Our findings align with those of [26], who observed no variation in basal serum cortisol levels between healthy individuals and convalescent patients 3 months after experiencing the initial symptoms of COVID-19. Notably, [26] reported lower cortisol levels in those patients affected by respiratory symptoms 3 months after the onset of COVID-19. Again, these data have identified a specific clinical subset of patients who may benefit from cortisone therapeutic management. The physiological relevance of how cortisol levels could be affected without dysfunction in the supra-renal axis needs further examination.

Factors identified as predictive of persistent disease include Epstein–Barr virus viremia and type 2 diabetes [26]. Moreover, individuals with Long COVID may have reactivated immune responses against EBV [25]. In our cohort, all participants underwent a positive anti-EBV test, and although type 2 diabetes was more commonly diagnosed in patients with Long COVID, it was not statistically significant, nor was the association of Long COVID with acute disease severity [42,43].

Whether the breakthrough SARS-CoV-2 infection in vaccinated people results in post-acute sequelae is not clear. In our cohort, the majority of enrolled participants received one shot of anti-SARS-CoV-2 vaccination. Other strong predictors of Long COVID included elevated antibodies against EBV and reduced levels of certain immune cells, as indicated above [25]. While our data align with some immune cell population alterations, we did not identify a causal role for adrenal insufficiency in Long COVID. On the other hand, our study found that participants with Long COVID had comparable levels of antibody positivity to EBV or CMV, as recovered participants, suggesting that differences in immune cell populations could be specifically attributed to post-SARS-CoV-2 infection.

Limitations: our study provides a preliminary glimpse into the immunological and neuroendocrine landscape of Long COVID. However, it is essential to acknowledge the limitations of this pilot study, including its small sample size and the need for a more extensive, multi-centre study with a diverse patient population. The limited number of enrolled patients does not allow for in-depth analysis of the association of lower cortisol levels and clinical symptoms or whether vaccination has an effect on Long COVID prevention. Moreover, a longitudinal study would be needed in order to explore whether Long COVID symptoms are tied to adrenal insufficiency and how humans recover over time. Including patients from different waves will allow us to explore how different variants affect Long COVID. Finally, as discussed above, the ACTH stimulation test with lower doses of ACTH will disclose adrenal insufficiency more accurately.

## 5. Conclusions

Our findings suggest that Long COVID is associated with specific immunological alterations, such as changes in B cell maturity, the presence of resident memory CD8+ T cells, and Treg cell dysregulation. These findings lay the groundwork for future research aiming to unravel the mechanisms underlying Long COVID and develop targeted therapies, or for interventions aiming to alleviate its symptoms and improve the quality of life for affected individuals. Further investigations are required to validate and expand upon these initial observations and to explore potential therapeutic avenues.

## Figures and Tables

**Figure 1 biomedicines-12-00581-f001:**
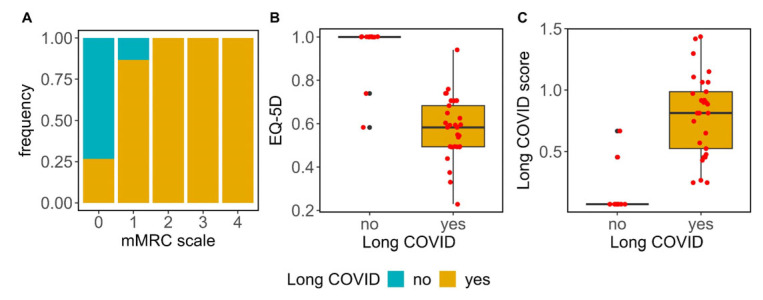
Quality of life and breathless affectation in Long COVID study participants. (**A**) Distribution of mMRC scores in patients with Long COVID and recovered participants. The figure classifies patients based on their mMRC scores. The recovered patients predominantly fall into the lower end of the mMRC scale, indicating minimal breathlessness. In contrast, Long COVID patients display higher mMRC scoring, suggesting more pronounced breathlessness during activity. Treating mMRC as an ordinal variable, the Fisher’s statistics test indicated a significant difference between the Long COVID and recovered participants, with a *p*-value < 0.001. (**B**) EQ-5D scores in Long COVID. The box plot illustrated that Long COVID patients displayed lower EQ-5D scores, indicating poorer quality of life than recovered patients. The difference is statistically significant with a *p*-value of 2.913 × 10^−6^ (Mann–Whitney statistics test). (**C**) Long COVID score (LCS). The box plot illustrates that Long COVID patients displayed higher LCS, computed by combining mMRC and EQ-5D, indicating both more breathlessness and poorer quality of life than recovered patients. Red dots are patients and black dots refers to those patients that fall significantly outside the typical range of values The difference is statistically significant with a *p*-value of 4.0087 × 10^−9^ (Mann–Whitney statistics test).

**Figure 2 biomedicines-12-00581-f002:**
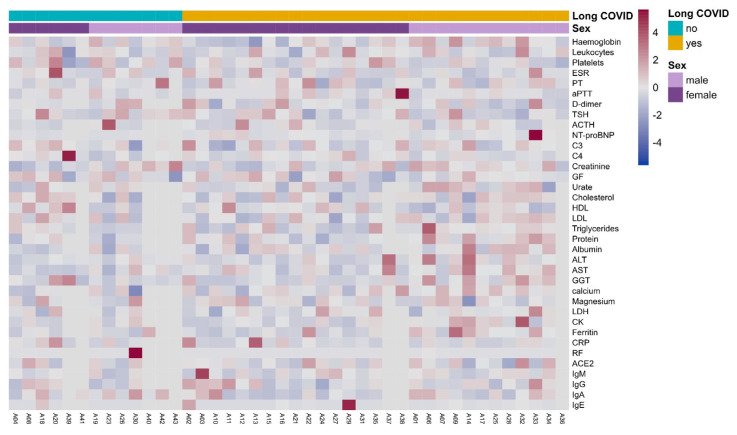
Clinical laboratory parameters in Long COVID cohort participants. Clinical parameters were measured during the medical visit to diagnose patient classification, either as having or not having Long COVID. A heatmap is plotted using the distribution of participants in columns depending on their diagnosis of Long COVID and sex. No clustering of clinical parameters was observed based on these two categories. ESR, erythrocyte sedimentation rate; PT, prothrombin time; aPTT, activated partial thromboplastin time; TSH, thyrotropin; NT-proBNP, N-terminal pro-B type natriuretic peptide; C3, complement component 3; C4, complement component 4; GF, glomerular filtrate; LDL, low-density lipoproteins; HDL, high-density lipoproteins; ALT, alanine amino transferase; AST, aspartate amino transferase; GGT, gamma-glutamyl transferase; LDH, lactate dehydrogenase; CK creatine kinase; CRP, C-reactive protein; RF, rheumatoid factor; ACE2, angiotensin converting enzyme-2; Ig, immunoglobulin. Mann–Whitney U test showed no association of any clinical laboratory variable with Long COVID.

**Figure 3 biomedicines-12-00581-f003:**
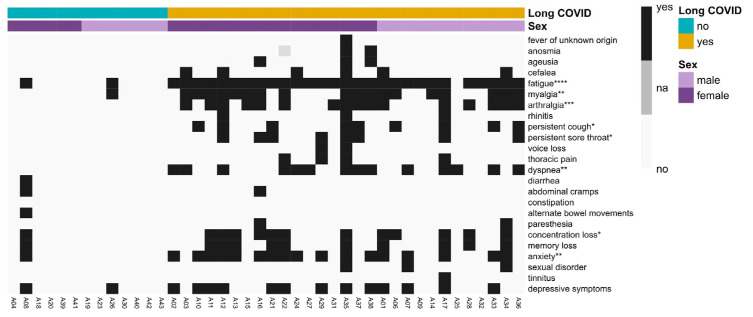
Presence of Long COVID symptoms in our Long COVID cohort. Long COVID symptoms were assessed as present (1, yes) or not (0, no) for all participants in our cohort and plotted in a heatmap. Participants were distributed along the columns based on their diagnosis of Long COVID and sex. Clinical symptoms showing a significant association with patients with Long COVID are indicated by asterisks, referring to their *p*-value significance: * *p*< 0.05, ** *p* < 0.01, *** *p* < 0.001, **** *p* < 0.0001. Fisher’s exact test was used.

**Figure 4 biomedicines-12-00581-f004:**
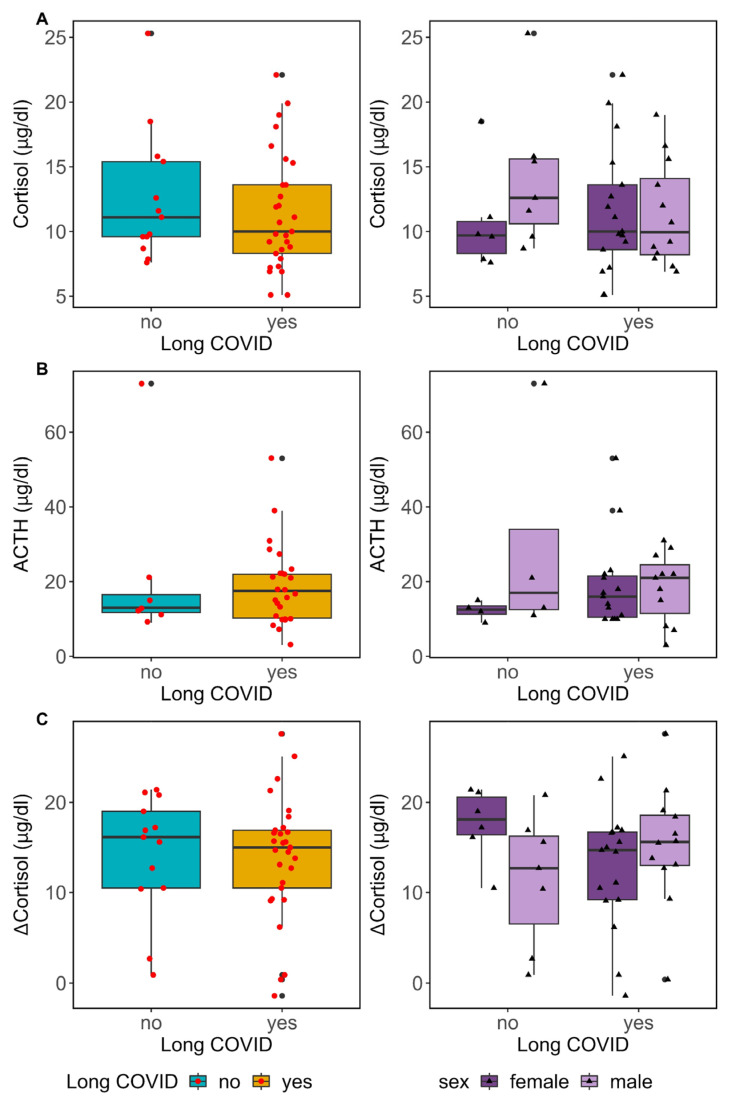
Pituitary–adrenal examination in our Long COVID cohort. Evaluation of basal cortisol and ACTH levels in the context of Long COVID. (**A**) Cortisol levels were measured at 8:00 a.m. for all participants and the results are subsequently presented in a box plot based on their Long COVID status. Left, the median levels of cortisol were equivalent in both groups (Mann–Whitney statistics test, *p*-value = 0.52). Right, participants with and without Long COVID were classified by sex, and cortisol levels were plotted for each group. The 2-way ANOVA statistical test showed no differences in any category (adjusted *p*-value: by Long COVID = 0.48; by sex = 0.54; interaction = 0.26). (**B**) ACTH levels were measured concomitantly to establish cortisol levels and the results are presented in a box plot according to Long COVID status. The median levels of ACTH were equivalent in both groups (*p*-value = 0.53) (Left), even when they were also analysed by sex (adjusted *p*-value: by Long COVID = 0.71; by sex = 0.45; interaction = 0.11). (**C**) Cortisol increases after 1 h of ACTH stimulation (ΔCortisol) were plotted based on the Long COVID status of participants (left) and also their sex (right). The 2-way ANOVA statistical test showed no differences in any category (adjusted *p*-value: by Long COVID = 0.29; by sex = 0.78; interaction = 0.67).

**Figure 5 biomedicines-12-00581-f005:**
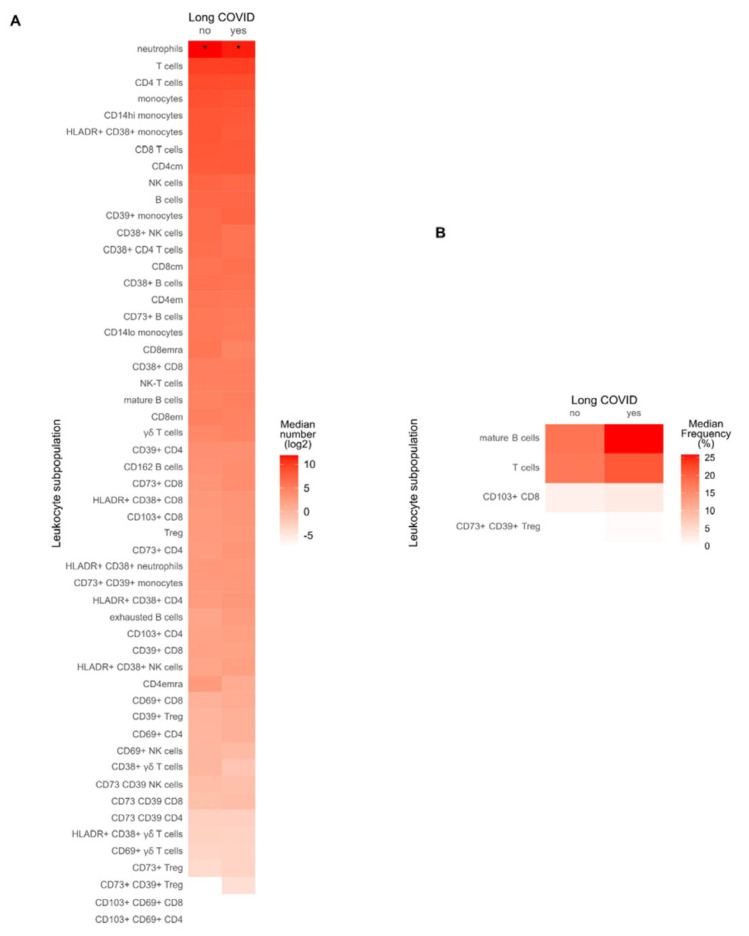
Blood immune cell predominance in the context of Long COVID. (**A**) Immune cell counts of different leukocyte subpopulations between patients with Long COVID (yes Long COVID) and recovered participants (no Long COVID) were plotted as the log2 of the median’s group. Asterisks point out those leukocyte subpopulations with significant differences between both groups. Mann–Whitney U test, * *p* < 0.05. (**B**) Leukocytes subpopulations with significant (*p* < 0,05) differences in their percentages between participants with or without Long COVID. The frequency referred to their parental cells (see Appendix A). cm, central memory; em, effector memory; emra, effector memory CD45RA-positive; NK, natural killer; Treg, regulatory T cells. The Mann–Whitney U test was performed.

**Figure 6 biomedicines-12-00581-f006:**
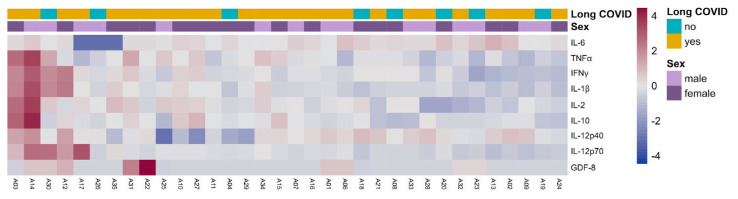
Inflammation in Long COVID. Heatmap plot of inflammatory molecules in participants with or without Long COVID, segregated by sex. Two-way ANOVA statistical test showed no differences in any category for each cytokine.

**Table 1 biomedicines-12-00581-t001:** Comparison of demographic, preclinical, and acute phase data between Long COVID and recovered patients.

	Long COVID		
**Variables**	**No (13)**	**Yes (29)**	***p*-Value ^1^**
Age (years)	50 [8]	53 [18]	0.64
Gender (female)	6 (46.15)	17 (58.62)	0.52
HTA	1 (7.69)	5 (17.24)	0.65
Dyslipidaemia	1 (7.69)	5 (17.24)	0.65
DM2	0 (0)	5 (17.24)	0.30
Previous smoker	3 (23.08)	5 (17.24)	0.69
BMI > 30	1 (7.69)	2 (6.9)	1.00
Lung disease	1 (7.69)	1 (3.45)	0.53
AIDs	0 (0)	2 (6.9)	1.00
Chronic fatigue	0 (0)	1 (3.45)	1.00
Fibromyalgia	0 (0)	1 (3.45)	1.00
Chemical sensitivity	0 (0)	1 (3.45)	1.00
Complications in acute phase			
Hospitalization	7 (53.85)	17 (58.62)	1.00
Bacterial lung coinfection	2 (15.38)	1 (3.45)	0.22
Respiratory bacterial sepsis	1 (7.69)	1 (3.45)	0.53
Other bacterial infection	0 (0)	1 (3.57)	1.00
Pulmonary thromboembolism	1 (7.69)	2 (6.9)	1.00
Treatment			
No treatment	0 (0)	2 (6.9)	1.00
Paracetamol	1 (7.69)	3 (10.34)	1.00
NSAIDs	1 (7.69)	7 (24.14)	0.40
Dexamethasone	7 (53.85)	16 (55.17)	1.00
Heparin	7 (53.85)	17 (58.62)	1.00
Tociluzumab	1 (7.69)	7 (24.14)	0.40
Remdesivir	0 (0)	2 (6.9)	1.00
Antibiotics	2 (15.38)	2 (6.9)	0.58

Data are *n* (%), but age is median [interquartile range (IQR)]. HTA, hypertension ≥ 160 mm Hg; dyslipidaemia, total cholesterol ≥ 200 mg/dL; DM2, diabetes mellitus type 2; BMI, body mass index; AIDs, autoimmune diseases; NSAIDs, non-steroidal anti-inflammatory drugs. ^1^ Fisher’s exact test used for all data but Mann–Whitney test applied for age.

## Data Availability

Research data generated during this study are saved in our institutional repository of Vall d’Hebron Institut de Recerca and will be available upon reasonable request to the corresponding authors.

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
