# Peer review of "Pituitary–Adrenal Axis and Peripheral Immune Cell Profile in Long COVID"

_biomedicines, 2024, doi:10.3390/biomedicines12030581_

Round 1

Reviewer 1 Report

Comments and Suggestions for Authors

This is an ambitious study aiming to compare the inflammatory and immune function as well as pituitary-adrenal axis in patients with long COVID syndrome compared to those who recovered from COVID infection without sequelae.

The extent of the laboratory investigation is very impressive but the study sample is small. More importantly, the diagnostic criteria for long COVID syndrome are very vague and lack standardization, severely limiting the interpretation of the results in my opinion.

The scales used in the figures to assess the clinical laboratory parameter are not defined and difficult to understand.

For the correct assessment of the pituitary-adrenal axis it is essential to know whether in the acute or postCOVID phase any glucocorticoids were used in any of the patients. Also, basal morning cortisol and ACTH levels have normal fluctuations and do not provide reliable conclusions regarding the optimal functionality of this axis.

Comments on the Quality of English Language

Minor corrections needed

Author Response

We thank the reviewers for the comments which we addressed one by one and believe that they are very helpful in improving the manuscript.

Reviewer #1

Q1: This is an ambitious study aiming to compare the inflammatory and immune function as well as pituitary-adrenal axis in patients with long COVID syndrome compared to those who recovered from COVID infection without sequelae.

A1: We thank the reviewer for this comment that exactly profiles our work.

Q2: The extent of the laboratory investigation is very impressive but the study sample is small. More importantly, the diagnostic criteria for long COVID syndrome are very vague and lack standardization, severely limiting the interpretation of the results in my opinion.

A2: The reviewer is absolutely right to make this remark and we agree with him regarding the fact that the cohort was quite small. However, we would like to clarify that in order to control all the parameters and to ensure that they were as uniform as possible, we designed a single-centre study with patients coming from a specific wave of infection that corresponded the vast majority of them around winter 2021. With regard to the diagnostic criteria for Long COVID, three expert internal medicine doctors carried out the medical visit and clinically assessed the patient's Long COVID status together with the test Post-COVID Functional Status (PCFS) that the patients completed. Once this classification had been made, it was corroborated whether the assignment of patients to the Long COVID group was in line with the health and quality of life tests (EQ-5D and mMRC). The description of this methodology can be found in sections 2.1 Inclusion Criteria for Patients, and 2.2 Health-Related Quality of Life Tests in the Materials and Methods section.

Q3: The scales used in the figures to assess the clinical laboratory parameter are not defined and difficult to understand.

A3: The figures are heatmap and data are presented as z-score where the values are normalized values. We believe that this type of figure shows in a more visual way the differences not only between groups of patients (Long COVID or not, and sex) but of each individual patient. Note that in each Heatmap figure, patients are identified by a code that is the same in all figures, and hence clinical laboratory data for the same patient could be linked. Importantly, the clinical values of the parameters measured in the laboratory are detailed in the tables attached as supplementary material and are indicated in the text of the manuscript next to each figure presented as a heatmap.

Q4: For the correct assessment of the pituitary-adrenal axis it is essential to know whether in the acute or postCOVID phase any glucocorticoids were used in any of the patients. Also, basal morning cortisol and ACTH levels have normal fluctuations and do not provide reliable conclusions regarding the optimal functionality of this axis.

A4: We fully agree with this observation. That is why in Table 1 we analysed in the acute phase that patients with Long COVID do not have a different therapeutic treatment from patients without Long COVID, in order to rule out any effect of treatment on the levels of the hormones ACTH and cortisol. As requested, cortisol levels at the time of Long COVID assessment were not different between patients who received dexamethasone and those who did not receive dexamethasone in the acute phase of COVID-19 (median [iqr], 9.7 [4.7] µg/dL vs 11.9 [6.7] µg/dL, respectively, p-value = 0.2155).

None of these patients were treated with glucocorticoids during the Long COVID phase of the study. This statement is now added to the new corrected version of the manuscript in the section 2.1 Inclusion Criteria for Patients.

Due to the large variation that these hormones can present depending on circadian rhythm, they were measured at 8 a.m., which is within the time range recommended by clinical guidelines. All protocols recommended by clinical guidelines such as sleep, diet, and rest time before blood collection were followed.

Reviewer 2 Report

Comments and Suggestions for Authors

Summary: This manuscript examines pituitary-adrenal axis function, immune cell profiles, and inflammation in patients with long COVID compared to recovered individuals. The key findings were:

Patients with long COVID had significantly worse quality of life and dyspnea scores compared to recovered individuals. No significant differences were found in basal cortisol or ACTH levels between groups. The ACTH stimulation test also showed no significant differences. Patients with long COVID had lower neutrophil counts but a higher percentage of mature B cells and resident memory CD8+ T cells compared to recovered individuals. No significant differences in inflammatory cytokines were found between groups. Overall, this is an interesting preliminary study on long COVID pathology. The findings on immune cell populations are novel and warrant further confirmation and characterization in larger studies. The manuscript would be strengthened by addressing the limitations discussed above.

Suggestions:

Provide more details on the patient demographics and pre-existing conditions. Clarify the timeline of COVID-19 vaccination in relation to primary infection and long COVID diagnosis. Include statistical test values and confidence intervals for key results. Carefully proofread the manuscript to fix minor grammar, style and formatting issues.

Author Response

We thank the reviewers for the comments which we addressed one by one and believe that they are very helpful in improving the manuscript.

Reviewer #2

Summary: This manuscript examines pituitary-adrenal axis function, immune cell profiles, and inflammation in patients with long COVID compared to recovered individuals. The key findings were:

  • Patients with long COVID had significantly worse quality of life and dyspnea scores compared to recovered individuals.
  • No significant differences were found in basal cortisol or ACTH levels between groups. The ACTH stimulation test also showed no significant differences.
  • Patients with long COVID had lower neutrophil counts but a higher percentage of mature B cells and resident memory CD8+ T cells compared to recovered individuals.
  • No significant differences in inflammatory cytokines were found between groups.
  • Overall, this is an interesting preliminary study on long COVID pathology. The findings on immune cell populations are novel and warrant further confirmation and characterization in larger studies. The manuscript would be strengthened by addressing the limitations discussed above.

Answer: we thank the reviewer for this accurate summary of the study.

Suggestions:

Q1: •Provide more details on the patient demographics and pre-existing conditions.

A1: Table 1 presents the demographic and clinical data of the patients included in the study. Six of the authors of this study are physician internists and all the variables they required or considered for this study were already included in Table 1. Please clarify if the reviewer needs any additional information.

Q2: •Clarify the timeline of COVID-19 vaccination in relation to primary infection and long COVID diagnosis.

A2: The time of primary infection for the enrolled patients lasted from December 2020 to July 2021. Vaccination was rolled out starting January 2021 with prioritisation criteria in place, and the general population was not vaccinated until summer 2021. Consequently, none of the enrolled patient was vaccinated before primary infection. The description of this context is already present in the manuscript (section 3.1 and section 3.4). The manuscript text indicates “Our enrolled participants received their first shot, on average, 153 [31] days after their primary infection”.

We also stated in the section 3.1 Long COVID Assessment: “The assessment of Long COVID took place, on average, 286 days after the primary infection, with a range of 217 to 346 days”.

It means that vaccination was done on average 133 days before Long COVID diagnosis. And as indicated in the manuscript, 88.2 % of patients had already received a first shot at the time of the Long COVID diagnosis.

Q3: •Include statistical test values and confidence intervals for key results.

A3: All continuous data results, including statistically significant and non-significant, include median and interquartile range, as well as the type of statistical test performed. Since we checked whether data were normalized or not and found that data from our cohort followed a non-Gaussian distribution, non-parametric statistics were applied.

Q4: •Carefully proofread the manuscript to fix minor grammar, style and formatting issues

A4: The entire manuscript has been carefully checked for grammatical, stylistic and formatting errors in order to correct them all. We highlighted the changes in the manuscript.

Reviewer 3 Report

Comments and Suggestions for Authors

Dear Editor

The manuscript “Pituitary-Adrenal Axis and Peripheral Immune Cell Profile in Long COVID” is well conducted study, that described some immune markers insights in Long COVID patients.

The introduction section consisted of unnecessary long descriptions to construct the hypothesis of the study. In routine, this sort of explanation is used in “Discussion”. The Methods section is very well described including inclusion criteria. Results are defined in detail along with description of data using Tables and Figures. The results and discussion section could be reduced.

Thanks and Regards

Author Response

We thank the reviewer for the comments which we addressed

Reviewer #3

Q1: The introduction section consisted of unnecessary long descriptions to construct the hypothesis of the study. In routine, this sort of explanation is used in “Discussion”. The Methods section is very well described including inclusion criteria. Results are defined in detail along with description of data using Tables and Figures. The results and discussion section could be reduced.

A1: We appreciate the reviewer's assessment and in order to remove all those parts that are considered unnecessary or not strictly required for the presentation of the manuscript, we have undertaken a re-reading of the introduction, the results and the discussion; and have eliminated all that we have considered not essential. We hope that the changes will be to the reviewer's liking. All these changes can be found in the manuscript as crossed out and highlighted in yellow.

Round 2

Reviewer 1 Report

Comments and Suggestions for Authors

I appreciate your feedback. However, no significant changes have been made to the manuscript, I maintain my initial evaluation

Author Response

We appreciate the reviewer for their evaluation. However, In this second review we do not see what are the points to be reviewed. We already addressed point by point the questions requested in the firts revision.